# NATURALPROVER: Grounded Mathematical Proof Generation with Language Models

Sean Welleck[1,2*], Jiacheng Liu[1*], Ximing Lu[2], Hannaneh Hajishirzi[1,2], Yejin Choi[1,2]
[1]Paul G. Allen School of Computer Science & Engineering, University of Washington
[2]Allen Institute for Artificial Intelligence, *Equal contribution
wellecks@uw.edu

## Abstract

Theorem proving in natural mathematical language – the mixture of symbolic and natural language used by humans – plays a central role in mathematical advances and education, and tests aspects of reasoning that are core to intelligence. Yet it has remained underexplored with modern generative models. We study large-scale language models on two new generation tasks: suggesting the next step in a mathematical proof, and full proof generation. We develop NATURALPROVER, a language model that generates proofs by conditioning on background references (e.g. theorems and definitions that are either retrieved or human-provided), and optionally enforces their presence with constrained decoding. On theorems from the NATURALPROOFS benchmark, NATURALPROVER improves the quality of next-step suggestions and generated proofs over fine-tuned GPT-3, according to human evaluations from university-level mathematics students. NATURALPROVER is capable of proving some theorems that require short (2-6 step) proofs, and providing next-step suggestions that are rated as correct and useful over 40% of the time, which is to our knowledge the first demonstration of these capabilities using neural language models.[1]

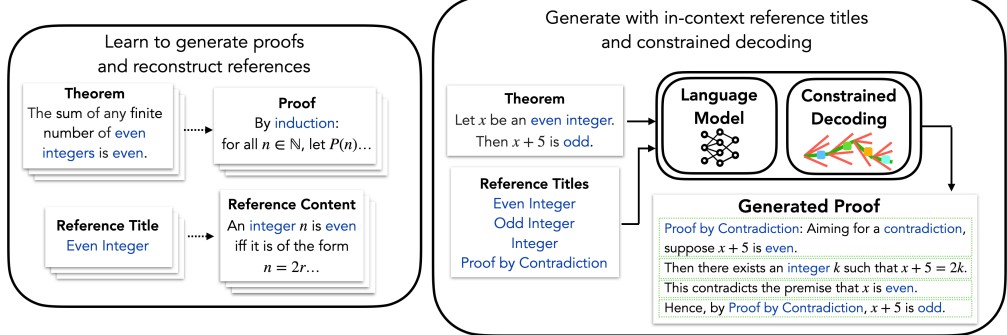

Figure 1: NATURALPROVER proves Even Integer Plus 5 is Odd. At training time, NATURALPROVER obtains background knowledge about references (e.g. theorems or definitions) via *reference reconstruction*: learning to map a reference's title to its content. At test time, NATURALPROVER grounds its generations through in-context reference constraints that are retrieved or human-provided, and optionally enforced with *stepwise constrained decoding*. This theorem's human-written proof in ProofWiki contains an error and differs substantially from NATURALPROVER's correct proof.

---

[1]Code and data available at `https://github.com/wellecks/naturalprover`.

36th Conference on Neural Information Processing Systems (NeurIPS 2022).

# 1   Introduction

Constructing a rational argument that justifies a claim is a key aspect of explaining, verifying, and communicating ideas in situations ranging from everyday interactions, to legal and political discourse, to science and mathematics [10, 42, 24]. Within the latter context, a *mathematical proof* – a sequence of logical arguments expressed in a mixture of symbolic and natural language – assumes this role by providing justification and insight into why a claim is true [12]. Proofs operate on a relatively explicit and objective set of ground knowledge, isolating a subset of reasoning that is desirable for models that form the foundation of machine learning systems [3]. Moreover, we envision assistive systems that provide suggested proofs or next-steps, analogous to language-model-based code suggestions (e.g. GitHub CoPilot [6]) or formal proof assistants (e.g. GPT-*f* [20]), which could make learning or using mathematics more productive and accessible.

To this end, we study the capabilities of large-scale language models (e.g. GPT-3 [5]) on two new theorem proving tasks in natural mathematical language: *next-step suggestion*, in which a model suggests the next step of a proof, and *full-proof generation*, in which a model fully proves a claim. As proofs are grounded in knowledge from past results (e.g. theorems, definitions), analogous to facts deployed in a conversation [13], prior rulings used in a legal opinion [16], or articles used to justify an answer [30], we develop a methodology for obtaining and using background knowledge to prove theorems with a generic language model.

We develop NATURALPROVER, a language model that generates proofs by conditioning on background references (e.g. theorems and definitions that are either retrieved or human-provided), and optionally enforces their presence with a constrained decoding algorithm that leverages the multi-step structure of proofs. On a collection of theorems from the NATURALPROOFS benchmark [45], NATURALPROVER improves the quality of next-step suggestions and generated proofs over fine-tuned GPT-3, according to human evaluations from university-level mathematics students. NATURALPROVER is capable of proving some theorems that require short (2-6 step) proofs, and providing next-step suggestions that are rated as correct and useful more than 40% of the time, which is to our knowledge the first demonstration of these capabilities using neural language models.

Along with these successes, we study deficiencies in our current models. We find that models can struggle with logical coherence on longer proofs, with providing valid justifications, and with performing multi-step symbolic derivations. Taken together, our tasks, methodology, and evaluation show the feasibility of language models as interactive aids in mathematics, along with open challenges.

# 2   NATURALPROOFS-GEN Dataset and Tasks

We create a NATURALPROOFS-GEN dataset adapted from NATURALPROOFS [45], and use the dataset for two tasks: suggesting the next step of a proof, and fully proving a theorem.

**NATURALPROOFS-GEN.** NATURALPROOFS-GEN adapts data from NATURALPROOFS, which contains theorem statements, proofs, definitions, and additional pages (e.g. axioms, corollaries) sourced from ProofWiki, an online compendium of community-contributed mathematical proofs. In NATURALPROOFS-GEN, each example $(\mathbf{x}, \mathbf{y}) \in \mathcal{D}$ pairs a theorem $\mathbf{x}$ with a gold proof $\mathbf{y}$, both of which are a mixture of text and LaTeX. [45] split the examples and reference sets into training, dev, and test splits to ensure that no theorem in the dev or test splits was mentioned in the training split. We adopt these splits of roughly 12.5k training, 1k validation, and 1k test examples, and sampled *core evaluation sets* with 100 dev and 100 test theorems that are used for human evaluation. The proofs contain additional structure, discussed next.

**Multi-step proof structure.** Each proof has a *multi-step* structure, meaning that a proof $\mathbf{y} = (y_1, \ldots, y_{|\mathbf{y}|})$ is a variable-length token sequence that is segmented into *proof steps*, where each step $y_t$ is itself a variable-length sequence of tokens (either text or Latex). The segmentation is largely determined by ProofWiki's formatting and community standards for structuring proofs, and we additionally merge steps to ensure that each step contains non-trivial semantic content. For example, Figure 1 shows a 4-step (generated) proof with each step highlighted in green.

**References.** Each proof mentions a variable-number of *references* $\{\mathbf{r}_1, \ldots, \mathbf{r}_{R_y}\}$ from a set $\mathcal{R}$ of roughly 33k theorems and definitions, analogous to how Wikipedia articles reference other pages. For example, Figure 1 shows a proof with reference mentions in blue. Each mention identifies a

reference by its title and provides a natural language surface form. For instance, in Figure 1, the first proof step mentions the definition of even integer as even, which is formatted in the proof as `[[Definition:Even_Integer|even]]` and tokenized along with the rest of the proof.

**Tasks.** We consider two tasks that are motivated by an assistive system that provides suggested proofs or next-steps to a user. The **full proof generation** task is to generate a proof $\mathbf{y}$ given a theorem $\mathbf{x}$. The **next-step suggestion** task is to generate a set of next steps $\{y_t^k\}_{k=1}^K$ given theorem $\mathbf{x}$ and proof history $y_{<t}$ from a gold proof. In each case, we consider an additional **provided reference** setting where the model is also given the set of references $\{\mathbf{r}_1^*, \ldots, \mathbf{r}_{R_y}^*\}$ from a gold proof of the theorem. The next-step task simulates a human correctly proving the theorem up to a point, then querying a system for suggested next-steps when stuck, while the provided reference setting simulates a human specifying a plan for a system that writes a proof.

## 3 NATURALPROVER: Grounded Proof Generation via Language Modeling

We describe NATURALPROVER, a language model which generates grounded proofs by conditioning on references and optionally enforcing their presence with constrained decoding.

**Setup.** Our objective is to generate correct proofs, $\hat{\mathbf{y}} = \arg\max_{\mathbf{y}} \text{correct}(\mathbf{x}, \mathbf{y})$. Unfortunately, evaluating proof correctness is costly, and is only done once at test time. A naive approach is to approximate the objective, $\hat{\mathbf{y}} \approx \arg\max_{\mathbf{y}} \log p_\theta(\mathbf{y}|\mathbf{x})$, by fine-tuning a language model $p_\theta$ on $(\mathbf{x}, \mathbf{y})$ examples and using a decoding algorithm (e.g. greedy decoding). We instead investigate conditioning on background knowledge in the form of reference documents, $p_\theta(\mathbf{y}|\mathbf{x}, R)$, which is beneficial in related generation settings (e.g. [38]), and offers control over the generated proof. To do so, NATURALPROVER uses in-context references and a reference reconstruction objective.

**In-context references.** Language models have a limited context window that prevents conditioning on full documents. Instead, NATURALPROVER conditions on a set of reference titles, $p_\theta(\mathbf{y}|\mathbf{x}, R_{\text{title}})$. Concretely, we fine-tune on (theorem, reference titles, proof) sequences of the form,

```
<theorem> <title> {theorem-title} </title> <content> {theorem-content} </content> </theorem>
```
```
<ref> {ref-title-1} </ref> ...  <ref> {ref-title-R} </ref>  <proof> {proof} </proof>
```
(1)

with new-lines and $\{\}$ tokens omitted, relevant strings inserted, and loss only on tokens after `<proof>`.

**Reference reconstruction.** Reference titles do not capture all of the information contained in the reference documents. We learn a mapping between each reference title and its underlying document with a reference reconstruction objective, $p_\theta(\mathbf{r}|\mathbf{r}_{\text{title}})$ for references $\mathbf{r}$ in the training reference set. Concretely, we fine-tune on additional (title, content) pairs of the form,

```
<{type}> <title> {title} </title> <content> {content} </content> </{type}>,
```
(2)

where the `{type}` is theorem/definition/other, and the loss is only on tokens after `<content>`. Intuitively, this lets the model associate each reference title with the reference's underlying content.

**The joint objective.** For training, we minimize the joint loss,

$$\mathcal{L}(\theta) = \frac{1}{|\mathcal{D}^{\text{train}}| + |\mathcal{R}^{\text{train}}|}\Big[\sum_{(\mathbf{x},\mathbf{y})\in\mathcal{D}^{\text{train}}} -\log p_\theta(\mathbf{y}|\mathbf{x}, R_{\text{title}}) + \sum_{\mathbf{r}\in\mathcal{R}^{\text{train}}} -\log p_\theta(\mathbf{r}|\mathbf{r}_{\text{title}})\Big]. \quad (3)$$

**Evaluation-time references.** We consider two settings for evaluation-time references: (i) *retrieved* references, from a retrieval model $f(\mathbf{x}) \rightarrow \{\mathbf{r}_1, \ldots, \mathbf{r}_k\}$, and (ii) *human-provided* references from the ground-truth proof. The retrieval setting simulates a fully automated proof assistant, while the second simulates a human specifying a plan for an assistant that writes a proof, and acts as an upper bound for a retrieval system optimized to predict references in a ground-truth proof.

### 3.1 Stepwise constrained decoding

In the provided-reference setting, the conditioned references are known to be relevant to a correct proof. We hypothesize that explicitly encouraging generated proofs to contain the references will improve correctness, by placing lexical constraints on the reference-titles at decoding time,

$$\hat{\mathbf{y}} \approx \arg\max_{\mathbf{y}} \log p_\theta(\mathbf{y}|\mathbf{x}, R_{\text{title}}), \text{ subject to } \sum_{\mathbf{r}_{\text{title}}\in R_{\text{title}}} \mathbb{I}\left[\mathbf{r}_{\text{title}} \in \mathbf{y}\right] = |R_{\text{title}}|, \quad (4)$$

where $\mathbb{I}[\cdot]$ is an indicator function. To approximate this objective, we generate step-by-step by sampling multiple proof-step candidates, retaining those with high value (reference coverage and log-probability) in a beam, and continuing to the next step, which we call stepwise beam search.

**Value function.** The search supports any function of the proof-so-far, $v(y_{\leq t}) \to \mathbb{R}$. We use a value function that is a weighted combination of constraint satisfaction and log-probability,

$$v_\alpha(y_{\leq t}) = \alpha v_{\text{constraint}}(y_{\leq t}) + (1 - \alpha)v_{\text{LM}}(y_{\leq t}), \tag{5}$$

where $v_{\text{constraint}}(y_{\leq t})$ is the number of unique in-context reference-titles in $y_{\leq t}$, and $v_{\text{LM}}(y_{\leq t})$ is $\log p_\theta(y_{\leq t})$. We normalize each term by dividing by the maximum absolute value among candidates.

**Stepwise beam search.** The procedure generates a proof $\mathbf{y} = (y_1, \ldots, y_T)$ by iteratively sampling and pruning next-proof-step candidates $y_t$. Each iteration expands a size-$K$ beam of proofs-so-far, $S_{t-1} = \{y_{<t}^k\}_{k=1}^K$, by generating $N$ next-step candidates,

$$S'_t = \cup_{y_{<t} \in S_{t-1}} \big\{ (y_{<t} \circ y_t^n) \mid y_t^n \sim q(\cdot|y_{<t}, \mathbf{x}, R_{\text{title}}) \big\}_{n=1}^N, \tag{6}$$

where $q$ is a decoding algorithm (e.g. temperature sampling) and $\circ$ is concatenation. The next iteration's beam is formed by selecting the top scoring candidates, $S_t = \arg\text{top-K}_{y_{\leq t} \in S'_t} v_\alpha(y_{\leq t})$. When a proof in the beam terminates, it is not expanded further. The search ends when the beam consists of $K$ terminated proofs. The highest value proof is returned as the final output.

**Stepwise++.** We add two mechanisms for promoting exploration at each step. First, we expand each prefix in the beam (Eqn. 6) by sampling with multiple temperatures, $\{y_t^n \sim q_\tau(\cdot|y_{<t}, \mathbf{x}, R_{\text{title}}) \mid \tau \in \{\tau_i\}_{i=1}^m\}$, where $q_\tau$ is sampling with temperature $\tau$. This relaxes the commitment to a single temperature for all proof steps, balancing exploration (higher $\tau$) with exploitation (lower $\tau$).

Second, rather than selecting the top-K candidates, we select clusters based on different value weights: $S_t = \cup_{\alpha \in \{\alpha_j\}_{j=1}^\ell} \text{top}_{K'}(S_t^\alpha)$, where $S_t^\alpha$ is the set of candidates scored with $v_\alpha$, and $K' = K/\ell$. This interpolates between selecting steps based on likelihood (low $\alpha$) and constraint satisfaction (high $\alpha$).

**Full proof sampling and greedy decoding.** An alternative is to sample full proofs and select the best one according to the value function. This can be viewed as expansion (Eqn. 6) done at the full proof, rather than the step level. Moreover, greedy decoding corresponds to expanding only 1 candidate with temperature $\to 0$. We formalize this in §D as a segment-level search that contains stepwise++, full proof sampling, and greedy decoding as special cases.

## 4 Proof Evaluation

A proof's correctness is contingent on a variety of factors, including reasoning with past results, performing symbolic derivations, and altogether providing sufficient evidence that the claim is true. We design a human-evaluation schema that isolates these aspects at the proof-step level, along with a full-proof summary. Table 1 summarizes the schema, which we overview below.

**References.** First, proofs involve deploying statements from references, such as applying a definition or adapting it to fit the context. Deployments should be consistent with the reference, e.g. deploying the definition of even integer as '...by definition, $\exists k \in \mathbb{Z} : x = 2k$...', rather than '...$\exists k \in \mathbb{Z} : x = 2k + 1$', and are a common source of errors in student proofs [15].

Second, proofs use references as justification for steps of reasoning; for instance, Real Addition is Commutative provides justification for the statement $x + y = y + x$ where $x, y \in \mathbb{R}$, but not for $xy = yx$. This aspect is analogous to using an article to justify a claim (e.g. [30]). Finally, proofs should not hallucinate references, or 'beg the question' by self-referencing the current theorem.

**Equations.** Proofs contain a variety of multi-step derivations, ranging from simple arithmetic to more sophisticated derivations (e.g. see Table 17). A derivation should start with a valid equation given the surrounding context (e.g. $x + x = 2x$ in Table 1 versus $x + x = 3x$). Each subsequent step should be a valid derivation from the previous step, e.g. stating $= (2k + 6) - 1$ after $y = 2k + 5$.

**Other reasoning, language, & symbolic errors.** A proof should provide sufficient evidence that a claim is true to a human reader; it should not skip steps. Proof steps should make progress towards proving the goal; in particular, they should not repeat known conditions in the theorem or conclusions made in a prior step. Finally, our schema leaves room for any other reasoning errors, as well as symbol errors (e.g. undefined symbols) and language errors (e.g. incomplete statements).

| Error Type | Example |
|---|---|
| **Reasoning: Reference** | |
| Invalid Deployment | Since $x$ is an even integer, $\exists k \in \mathbb{Z} : x = 2k + 1$. |
| Invalid Justification | $\mathbb{E}(X^2) = \sum_{k=1}^{n} k^2 \Pr(X = k)$    Power Series for Exponential Function |
| Hallucinated Ref. | From Power of Number are Irrational , $\sqrt[3]{2}$ is irrational. |
| Self Loop | (Proving Pythagoras's Theorem:)    From Pythagoras's Theorem, $c^2 = a^2 + b^2$. |
| **Reasoning: Equation** | |
| Invalid Equation | $\forall x \in \mathbb{R}, x + x = 3x$. |
| Invalid Derivation | (Since $x$ is an even integer, $x + 1 = 2r + 1$)    $= 2(r + 1)$ |
| **Reasoning: Other** | |
| Skips Steps | ($x \in \mathbb{Z}$ is not a multiple of 3.)    Therefore, $x^3 \equiv 1$ or $8 \pmod 9$ |
| Repetition | (Let $\triangle ABC$ be a right triangle.)    Then $\triangle ABC$ is a right triangle. |
| Invalid (Other) | ($x$ is an even integer.)    So, $x + 1$ is an even integer. |
| **Language** | Let $c = \sqrt{a^2 \backslash \text{add } b^2}$ be the    *( incomplete statement ; unknown symbol $\backslash add$ )* |
| **Symbolic** | (Let $x \in \mathbb{R}$.)    Let $y = x \circ x^{-1}$.    *( undefined operator $\circ$ for real numbers )* |

Table 1: Overview of human evaluation error schema. See Table 24 for full schema. Reference. Hallucinated reference . The necessary context (e.g. known conditions, prior steps).

Usefulness and correctness. To judge the potential utility of language models as assistive systems in natural mathematics, we measure whether generated next-steps and full proofs are potentially useful hints for proving the theorem on one's own. Additionally, we measure a summary judgment of correctness. Note that an incorrect statement can still be helpful; for instance, it could give a hint for the type of reference to use, derivation to perform, argument to make, etc.

**Human evaluation protocol.** We measure these aspects through human annotation at a *step-wise* and an *overall* level. For a step-wise annotation, an annotator is presented with the theorem, proof-so-far, and a generated next-step. The annotator labels the $\{0, 1\}$ correctness, usefulness, and presence of fine-grained errors outlined above. After labeling each step of a proof, the annotator rates the full proof's overall correctness and usefulness on a 0-5 scale. A rating of 4 or 5 is needed to be considered as correct, and a rating of 3 or above is needed to be considered as useful.

**Automatic metrics: lexical content.** As automatic proxies for quality, we compare each generated proof against its ground-truth counterpart using the sentence-level $n$-gram matching metric GLEU [29], and following work in knowledge-grounded dialogue [38] we use F1 overlap between generated and ground-truth tokens. Prior to computing the metrics, we normalize the generated and ground-truth proofs by only keeping the surface form of references, removing formatting characters with a MediaWiki parser, and collapsing any consecutive whitespace into a single space.

**Automatic metrics: knowledge grounding.** We define knowledge grounding as meaning that a generated proof contains the same references as those found in the ground-truth proof. To measure this, we use precision, recall, and F1-score between the reference sets contained in the generated and ground-truth proofs; i.e. $m(\{\hat{\mathbf{r}}_1, \ldots, \hat{\mathbf{r}}_{\hat{R}}\}, \{\mathbf{r}_1^*, \ldots, \mathbf{r}_{R_*}^*\})$, where $m(\cdot)$ is precision, recall, or F1. We also use Knowledge Token-F1 (kF1) ([38]), the overlap of the generated proof's tokens with tokens contained in the references mentioned in the ground-truth proof.

# 5 Experiments

We use the training and dev splits of NATURALPROOFS-GEN during fine-tuning, and the *core evaluation sets* consisting of 100 theorems from the validation set and 100 from the test set for evaluation (see §2). These theorems were selected by the authors such that by looking at the theorem title each author could recall its content and sketch a proof. While this may shift the evaluation towards an easier slice of the dataset, it was necessary to make human evaluation at a meaningful scale feasible. We also use the core sets for explorations and ablations.

We finetune three GPT-3 [5] (Curie) models, using the OpenAI API (see Appendix E for details):

| | Reasoning Errs ($\downarrow$) | | | Lexical Errs ($\downarrow$) | | Per-Step ($\uparrow$) | | Full Proof ($\uparrow$) | |
|---|---|---|---|---|---|---|---|---|---|
| | Ref. | Eqn. | Other | Lang. | Sym. | Useful | Correct | Useful | Correct |
| GPT-3 | 30.92 | 32.54 | 40.15 | 5.61 | 5.24 | 25.69 | 28.18 | 20% | 13% |
| NATURALPROVER$_{\text{RETRIEVE}}$ | **23.52** | 37.55 | 23.66 | 4.54 | 6.19 | 41.54 | 33.56 | 32% | 24% |
| NATURALPROVER | 25.84 | 35.93 | 25.23 | 8.41 | 5.35 | 39.60 | 26.30 | 35% | 24% |
| NATURALPROVER$_{++}$ | 23.61 | **28.54** | **18.45** | 5.58 | 3.65 | **46.57** | **35.41** | **45%** | **32%** |
| Next-step (NATURALPROVER) | 19.70 | 26.32 | 19.10 | 8.57 | 5.86 | 51.43 | 42.86 | – | – |

Table 2: Human evaluation results on the core test set for full proof generation and next-step suggestion (bottom row). All models are fine-tuned on NATURALPROOFS-GEN. Knowledge – either retrieved or human provided – and constrained decoding improve proof generation, with 46% of proof steps rated as useful and 35% correct according to university-level mathematics students.

1. **Baseline GPT-3.** We finetune a baseline GPT-3 model, $p_\theta(\mathbf{y}|\mathbf{x})$, on theorem-proof examples $\{(\mathbf{x}, \mathbf{y})\}$ from the training split. At test time, we condition the model on a test theorem.

2. **NATURALPROVER$_{\text{RETRIEVE}}$.** We finetune GPT-3 with retrieved references, $p_\theta(\mathbf{y}|\mathbf{x}, \hat{\mathbf{r}}_1, \ldots, \hat{\mathbf{r}}_{20})$. We use a pretrained joint retrieval model $f(\mathbf{x}) \rightarrow (\mathbf{r}_1, \ldots, \mathbf{r}_{|\mathcal{R}|})$ from [45], which was trained to retrieve an input theorem's ground truth references. At test time, the model receives a theorem and the top-20 reference titles that are retrieved given the theorem.

3. **NATURALPROVER.** We finetune GPT-3 with human-provided references, $p_\theta(\mathbf{y}|\mathbf{x}, \mathbf{r}_1^*, \ldots, \mathbf{r}_{R_{\mathbf{y}}}^*)$, where $\{\mathbf{r}_1^*, \ldots, \mathbf{r}_{R_{\mathbf{y}}}^*\}$ is the set of reference-titles in the ground-truth proof. We use reference-title conditioned examples (Eqn. 1) and reference-reconstruction (Eqn. 2) on the training split/reference set. At test time, the model receives a theorem and reference titles from its ground-truth proof.

For **next-step suggestion** we use the human-provided knowledge model (NATURALPROVER).

**Decoding.** For full proof generation, we use stepwise++ decoding with the provided knowledge model, which we refer to as **NATURALPROVER$_{++}$**, and otherwise use greedy decoding. We do not use stepwise constrained decoding with retrieved references since these references introduce noisy constraints, nor for next-step prediction since the algorithm is designed for multi-step proofs. See §E for additional experimental details.

**Human evaluation setup.** To evaluate the proofs generated by NATURALPROVER, we recruited 15 students from the Department of Mathematics and Applied Mathematics at the University of Washington, including undergraduate, masters, and Ph.D. students. The annotators were trained on how to evaluate proof correctness and compensated according to IRB requirements; see §F.2. For each task, we first reveal the theorem and its gold proof to the annotator. If they cannot understand a theorem or its gold proof, they may skip evaluating it. Otherwise, they may proceed to see the model-generated proof, one step at a time, and annotate each step under the step-wise evaluation schema (outlined in §4). After all the steps are shown and evaluated, for the full-proof generation task, the annotator is asked to annotate the entire proof under the overall evaluation schema.

## 5.1 Main Results

**Our best method is capable of generating correct and useful proofs.** According to human evaluation results (Table 2), our best method is NATURALPROVER with human-provided references and stepwise++ inference. 32% of the proofs generated by this method are rated as correct and 45% are rated as useful as an aid for human proof writers. On the per-step level, 35% of the proof steps are correct and 47% are useful. Taking a more granular view, our best method makes significantly less reference, equation, and other reasoning errors than other baselines. It makes very few language and symbolic errors, meaning it produces mostly complete, well-formatted mathematical statements, and defines and uses symbols accordingly. It mostly avoids hallucinating references or creating circular proofs. On the other hand, despite improving over the other methods, the model often struggles with correctly deploying and using references (23.6% reference error rate), as well as symbolic computations (28.5% equation error rate), especially multi-step derivations (21.9% invalid).

**What do the model's correct proofs look like?** We inspected the proofs labeled as correct and found three main categories: (1) *reference-assembly* proofs whose correctness is heavily determined by reference statements (e.g. Table 18, Table 20); (2) *template-adaptation* proofs in which the model adapts the structure and content of a training theorem's proof to prove the unseen evaluation theorem

| Theorem | **Singleton Set is not Dense-in-itself** | |
|---|---|---|
| | Let $T = (S, \tau)$ be a topological space. | |
| | Let $x \in S$. | |
| | Then the singleton set $\{x\}$ is not dense-in-itself. | |
| GPT-3 | Let $x \in S$. | ✓ |
| | From Clopen Points in Topological Space we have that $\{x\}$ is open in $T$. | ✗ |
| | From Point is Open iff it is not Dense-in-itself, it follows that $\{x\}$ is not dense-in-itself. | ✗ |
| NATURALPROVER++ | From Singleton Point is Isolated, $\{x\}$ has an isolated point. | ✓ |
| | Hence the result by definition of dense-in-itself. | ✓ |

Singleton Point is Isolated
Let $T = (S, \tau)$ be a topological space.
Let $x \in S$.
Then $x$ is an isolated point of
   the singleton set $\{x\}$,
but not necessarily an isolated point of $T$.

Dense-in-itself
Let $T = S, \tau$ be a topological space.
Let $H \subseteq S$.
Then $H$ is dense-in-itself iff it contains no isolated points.

Table 3: GPT-3 hallucinates references, while the knowledge-grounded NATURALPROVER++ with constrained decoding correctly uses references, resulting in a correct and useful proof.

(e.g. Table 21, Table 22); (3) *complex* proofs that are not fully determined by reference statements and differ significantly from training proofs (e.g. Figure 1, Table 3). In terms of techniques, our method demonstrates some ability to produce direct proofs (Table 19), proofs by cases (Table 22), proofs by induction (Table 23), utilize references (Table 20) and do symbolic computations (Table 21).

**Vanilla fine-tuned GPT-3 struggles with proof generation.** The vanilla fine-tuned GPT-3 model yielded fewer useful and correct proofs, with more reference-based and other reasoning errors than all three knowledge-grounded settings. The model showed severe reference hallucination (18%) and repetition (23%). It also makes significantly more reasoning errors related to reference usage. Language and symbolic error rates roughly stay the same. Overall, naively fine-tuning GPT-3 on theorem-proof examples alone is suboptimal for proof generation.

**Human-provided knowledge improves proof generation.** Grounding the generations with human-provided references significantly raises correctness and usefulness of the proofs in both full-proof and per-step evaluation. It most substantially reduces reference errors, especially invalid deployments and hallucinated references. For example, Table 3 shows the model grounding a proof with information from the theorem Singleton Point is Isolated and the definition of Dense-in-itself, in contrast to the vanilla GPT-3 model which hallucinates references.

**Retrieved knowledge also improves proof generation.** Retrieved knowledge also turns out to be very helpful, and even comparable to human-provided knowledge in some metrics. Although the retrieval model is far from perfect, the proof generation model is capable of narrowing down the retrieved reference titles provided in its context, assembling proofs that are useful and correct more often than the no-knowledge model. Qualitatively, we found examples where grounding in retrieved references eliminates repetition, enables multi-step derivations justified by references (Table 21), and assembles references into a correct proof (Table 20). This paves a promising path towards fully automated mathematical proof generation in natural mathematical language.

**Constrained decoding further improves proof generation.** Table 4 confirms that stepwise++ decoding approximates the constrained objective (Eqn. 4) better than greedy search, yielding proofs with lower perplexity and higher constraint satisfaction (Ref-F1). This translates to generations that are correct and useful more often according to the annotators. Intuitively, the constraints encourage the model to include references that help prove the claim (e.g. Table 18).

| In-context | Stepwise++ | PPL ($\downarrow$) | Ref-F1 ($\uparrow$) |
|---|---|---|---|
| ✗ | ✗ | 1.0639 | 26.33 |
| ✗ | ✓ | 1.0549 | 30.07 |
| ✓ | ✗ | 1.0644 | 89.43 |
| ✓ | ✓ | 1.0549 | 94.25 |

Table 4: Stepwise++ decoding approximates the constrained objective better than greedy decoding, resulting in both lower perplexity and better reference coverage, regardless of whether knowledge is provided in-context.

**Next-step suggestion.** The next-step suggestion task characterizes a model's performance on making a single proof step given a correct proof-so-far. In Table 2 we use the provided-knowledge model with greedy decoding for next-step suggestion, and find that reasoning errors decrease and per-step usefulness and correctness improve compared to the full proof setting, with 51% of the proof steps rated as useful and 43% correct. Although we used a single suggestion in our human evaluation study, in Table 5 we simulate a user choosing from among multiple suggestions by sampling 10 next-steps from our model and computing automatic metrics on the sample with the best sum of metrics. Us-

| | Lexical | | Grounding | | | | |
|---|---|---|---|---|---|---|---|
| | GLEU | Token F1 | kF1 | Ref-P | Ref-R | Ref-F1 | Halluc (↓) |
| GPT-3 | 24.40 | 49.96 | 49.30 | 29.93 | 24.73 | 23.69 | 17.92 |
| NATURALPROVER$_{\text{RETRIEVE}}$ | 26.58 | 53.02 | 55.88 | 38.17 | 28.48 | 27.10 | 2.25 |
| NATURALPROVER | 35.27 | 66.00 | 90.07 | 93.05 | 86.05 | 87.08 | 1.60 |
| NATURALPROVER$_{++}$ | 34.49 | 65.61 | 96.39 | 94.66 | 95.00 | 93.92 | 1.71 |
| *human metric* | | | | | | | |
| Correctness [full] | 0.93 | 0.91 | 0.86 | 0.83 | 0.85 | 0.85 | 0.94 |
| Usefulness [full] | 0.90 | 0.87 | 0.82 | 0.78 | 0.80 | 0.80 | 0.97 |
| Correctness [step] | 0.81 | 0.80 | 0.74 | 0.69 | 0.73 | 0.72 | 0.97 |
| Usefulness [step] | 0.65 | 0.61 | 0.53 | 0.47 | 0.52 | 0.51 | 0.98 |
| Reasoning Errors: Ref. | 0.71 | 0.64 | 0.52 | 0.48 | 0.50 | 0.50 | 0.95 |
| Reasoning Errors: Eqn. | 0.70 | 0.74 | 0.75 | 0.69 | 0.74 | 0.73 | 0.78 |
| Reasoning Errors: Other | 0.65 | 0.61 | 0.53 | 0.47 | 0.52 | 0.51 | 0.98 |
| Language Errors | 0.99 | 1.00 | 0.99 | 0.98 | 0.99 | 0.99 | 0.73 |
| Symbolic Errors | -0.72 | -0.80 | -0.88 | -0.89 | -0.89 | -0.88 | -0.21 |

Table 6: Automatic metrics on the core test set for full-proof generation, and correlation between human metrics and automatic metrics on the core validation set.

ing 10 samples instead of greedily decoding a single sequence substantially improves each metric, suggesting that utility might be increased further by presenting multiple suggestions.

| Decoding | GLEU | Ref-F1 |
|---|---|---|
| Greedy | 47.87 | 65.50 |
| Temp (t=.6) | 60.60 | 84.44 |
| Temp (t=.8) | 61.89 | 86.74 |
| Temp (t=1.0) | **62.12** | **86.87** |

Table 5: *Next-step suggestion*: Sampling 10 suggestions improves over a single greedy suggestion.

**How good are Automatic Metrics?** We study how well the automatic lexical and grounding metrics introduced in (§4) can reflect the real quality of proofs, as a guide for using them as a proxy evaluation protocol for NATURALPROOFS-GEN. We compute the Pearson correlation coefficient between each pair of human and automatic metrics, with data from the four experiment settings for full-proof generation. Results are shown in the lower part of Table 6, with error metrics negated, meaning positive correlation is desired.

The lexical and grounding metrics positively correlate with full proof correctness and usefulness ($\geq 0.8$). At the step-level, the metrics show (i) high correlation with step-level correctness and language errors; (ii) varied, but positive, correlations with aggregate reasoning errors; (iii) negative correlation with symbolic errors (though symbolic errors are relatively low for all models). The results suggest that optimizing for automatic metrics may be a viable strategy, albeit without guarantees on how finer-grained reasoning aspects vary across proofs.

### 5.2 Ablations and error analysis.

| Recon. | Gleu | Ref-F1 | Halluc. |
|---|---|---|---|
| ✗ | 33.03 | 82.85 | 3.32 |
| ✓ | **35.93** | **84.15** | **2.68** |

Table 7: Effect of reference reconstruction in NATURALPROVER (greedy decoding, full validation set).

**Reference reconstruction.** We fine-tune an additional GPT-3 model that is provided with in-context reference titles, but without reference reconstruction. As seen in Table 7, reference reconstruction improves content and reference usage.

**Constrained decoding.** First, Table 9 compares the step-level search in stepwise++ with searching at the full-proof level through sampling multiple proofs and selecting the best with the NATURALPROVER value function (*rerank (n)*). Reranking 60 samples matches the cost of stepwise++ in terms of number of decoded tokens. Full-proof reranking yields the best Gleu, though with lower reference-F1. Second, Table 8 shows that the expansion and selection mechanisms together result in the best reference matching, while holding Gleu at a similar level. Finally, Table 13 shows that both terms in the NATURALPROVER value function $\alpha v_{\text{constraints}} + (1 - \alpha) v_{\text{LM}}$ are needed: increasing the constraint weight $\alpha$ increases reference-matching, with a tradeoff in Gleu at high values.

**Language model comparison.** Table 10 varies the language model used to parameterize NATURALPROVER . The content and reference usage metrics improve with larger models. Separately, we find that increasing inference-time compute closes the gap in reference-matching between GPT-2 and the larger GPT-3 model (Table 11): sampling 10 full-proofs from GPT-2 and selecting the best

| Expand | Select | GLEU | Ref-F1 |
|--------|--------|------|--------|
| ✗ | ✗ | 40.62 (.84) | 91.78 (.49) |
| ✓ | ✗ | 41.12 (.58) | 92.61 (.63) |
| ✗ | ✓ | 39.14 (.55) | 93.11 (.34) |
| ✓ | ✓ | 40.11 (1.55) | **94.13** (.45) |

| Decoding | Gleu | Ref-F1 |
|----------|------|--------|
| Greedy | 41.12 (–) | 89.30 (–) |
| Rerank (10) | **43.88** (.29) | 91.72 (.28) |
| Rerank (60) | 42.23 (.80) | 93.16 (.27) |
| Stepwise++ | 40.11 (1.55) | **94.13** (.45) |

Table 8: Ablation of the stepwise++ expansion and selection mechanisms. Mean (std) over 3 runs shown on the core dev set.

Table 9: Stepwise versus full-proof search. Mean (std) over 3 runs on the core dev set.

using the NATURALPROVER value function achieves the same reference-F1 as GPT-3 with a single greedily-decoded proof. However, Gleu remains much higher with the larger GPT-3 model.

**Challenge: Reasoning with references.** Although reference reasoning errors were decreased through knowledge-grounding and constrained decoding, NATURALPROVER still commits a reference error on 23.6% of test steps (27% dev), with 15% of steps containing invalid deployments and 10% invalid justifications. For next-step prediction, the reference error rate remains nontrivial (19.7% test, 13% dev). , meaning that the model can struggle to correctly deploy references or use them as justification even in the absence of compounding errors from previous steps. Table 15 shows example invalid deployments and justifications; the errors are at times subtle, and require reasoning about the theorem statement, reference content, and proof context.

**Challenge: Equations and derivations.** NATURALPROVER commits an equation-related error on 28.5% of test steps (22.8% dev), including invalid equations (9.4%) and derivations (21.9%). Though an improvement over vanilla fine-tuned GPT-3 (32.5%), the errors occur frequently and remain high for next-step prediction (26%). Table 17 shows representative errors, which range from simple 'commonsense' mistakes (e.g. $24 = 2^3$) to making invalid steps with false justification within more sophisticated multi-step proofs. Investigating the role of pretraining, in-context techniques [31], and autoformalization [39] is interesting future work.

**Challenge: Proof length.** Although NATURAL-PROVER demonstrates some ability to write long proofs (e.g. Table 23), the 42% next-step correctness suggests that compounding errors are likely as proof length increases. Indeed, our best model's full-proof correctness is 48% on 1-4 step proofs ($n = 102$), decreasing to 15.6% on proofs with 5 or more steps ($n = 64$), with lower per-step usefulness and correctness at later steps (Figure 2). Our findings are analogous to recent work on language modeling for formal theorem proving [32], where current models are typically limited to chaining 2 or 3 non-trivial steps of mathematical reasoning.

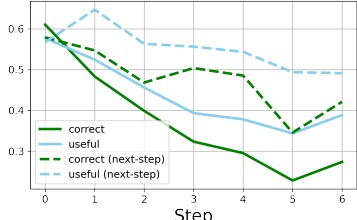

Figure 2: Per-step correctness and usefulness as a function of step number, for full-proof generation with NATURALPROVER++ and next-step prediction with NATURALPROVER.

## 5.3 Additional discussion

Finally, we provide higher-level comments on future work related to interactive systems, mathematical assistants, and generating proofs in informal versus formal mathematics.

**Interactive & improving systems.** Currently, our tasks are at two ends of a spectrum: in next-step generation, we always assume previous steps are from a human-written proof, while in full proof generation they are always from the model. Our results with multiple next-step suggestions suggest that users might find *some* suggestion among the multiple returned useful at a high rate, pointing to a middle ground: a human-in-the-loop NATURALPROVER, in which a human picks the next step from among the returned suggestions, or writes one based on the suggestions. The selected or written next-step could then be used as feedback to improve the system, enabling an iteratively improving NATURALPROVER. This notion of a continuously improving, teachable system is an emerging (e.g. [9]) and interesting future direction.

**Assistants for mathematics.** Our tasks were motivated by an assistant that helps a user write a proof, either from scratch or when stuck part of the way through. Our study here focuses on *capability*:

investigating whether neural language models are capable of performing the underlying mathematics that would be expected from such an assistant. A further challenge is to also ensure *reliability* – a user should have confidence that the model is not deceptive or incorrect, and is robust to changes in domain, on nearby problems, and on alternative ways of expressing a problem. Even further, we would like *flexibility* – human teachers can interact with a student flexibly through dialogue, natural language, and diagrams, rather than the strict input-output format defined by a dataset. Our work provides an initial step towards this larger vision.

**Informal and formalized mathematics.** Our work investigates theorem proving entirely in natural mathematical language (i.e. 'informal' mathematics), as it reflects an interface that a student typically uses when working with mathematics. An alternative is proving theorems in a formalized system, in which proof steps are expressed in a programming language (e.g. Lean [11]). Operating purely in a formalized system allows for verifying correctness – unlike our setting which must be verified by a human – arguably at the cost of flexibility and interpretability, as the mathematics is no longer expressed in natural language and must adhere to constraints of the formal system. Investigating combinations of the two – e.g. expressing a theorem in natural language, receiving a verified formal proof, then providing an interpretation in natural language – presents a wide range of interesting directions for future work.

# 6 Related Work

**Formalized mathematics with neural language models.** A large portion of work on machine learning for mathematics focuses on formalized mathematics. Language models have been used for interactive theorem proving, including in GPT-*f* [33, 32], PACT [20], and in [41]. In these settings proof steps are expressed in a programming language (e.g. Lean [11]) and there is access to a verifier, which differs from our setting of theorem proving in natural mathematical language.

**Informal mathematics with neural language models.** Previous work on theorem proving in natural mathematical language focuses on retrieving relevant premises (e.g. theorems, definitions) [17, 18, 45, 21], or informal-to-formal translation [43], which differ from our setting of generating next-steps or full proofs. Outside of theorem proving, various works use sequence models for problem solving, including benchmarking language models on arithmetic [37] or competition problems [22], symbolic mathematics [25, 46], augmenting LMs with verifiers [7] or in-context rationales [44] for math word problems, or using language models for math-related program synthesis [2, 14] and competitive programming [26]. These settings focus on generating executable programs or a numerical answer, which differ from our theorem proving setting, where the goal is to generate sound and convincing arguments on a range of topics in natural mathematical language.

**Related areas in NLP.** Systematic reasoning in natural language (outside of math) has been studied with synthetic proofs [36, 40], single-step deductions [4], or entailment trees [8], which differ from proving real-world mathematical theorems. Augmenting LMs with knowledge reduces hallucinations in dialogue [38] which has an analogous step-wise structure, while [30] use references within long-form answers; these and related NLP findings differ from improving the utility of mathematical proofs. Lexically-constrained decoding algorithms include variants of (token-level) beam search (e.g. [1, 23, 28, 27]) which assume access to per-token logits, and gradient-based decoding [34]; our segment-level decoding only assumes a sampler that returns text and its log-probability, making it compatible with recent language model API interfaces (e.g. the GPT-3 API).

# 7 Conclusion

We described NATURALPROVER, a knowledge-grounded language model that generates mathematical proofs by conditioning on background theorems and definitions, and optionally enforces their presence with constrained decoding. Our system improves the quality of next-step suggestions and generated proofs over fine-tuned GPT-3, demonstrating an ability to correctly prove theorems and provide useful suggestions to human proof writers.

## Acknowledgments and Disclosure of Funding

This work was funded in part by the Natural Sciences and Engineering Research Council of Canada (NSERC) (funding reference number 401233309), DARPA MCS program through NIWC Pacific (N66001-19-2-4031), and the Allen Institute for AI. We also thank Google Cloud Compute, as well as OpenAI.

The authors would like to thank Alisa Liu, Julian Michael, Yuren (Rock) Pang, and Kaiming Cheng for dogfooding and providing valuable feedback to our human evaluation system. We would also like to thank James McGivern for developing an interactive demo for NaturalProver.

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
