# OpenReview forum: "NaturalProver: Grounded Mathematical Proof Generation with Language Models"
_NeurIPS.cc/2022/Conference — NeurIPS 2022 Accept_

### Official Review · Reviewer_Cdcz · 2022-07-12

**Rating:** 7
**Confidence:** 4
**Soundness:** 3 good
**Presentation:** 4 excellent
**Contribution:** 3 good

**Summary:**

The authors use the NaturalProofs-Gen dataset to fine-tune GPT3 using two new loss functions: proof generation using in-context references and reference reconstruction. Additionally, they perform constrained decoding by adding lexical constraints to encourage generations that contain the input references. They evaluate the system on multiple metrics like the usefulness and correctness of the proof steps. Lastly, they compare the model's errors by human evaluation on sampled subsets of the eval dataset.

**Questions:**

* What is the order of the training data shown to GPT3 during fine-tuning - between in-context and reference reconstruction? Is it just randomly sampled or do you first show some reference reconstruction and then an in-context example that uses the previously shown references?

**Limitations:**

The authors clearly discuss the main limitations of the work, along with a comprehensive error analysis

**Strengths And Weaknesses:**

Strengths:
* Overall, the approach is very nicely presented and easy to follow through
* The experiments and ablations are comprehensive and well presented
* Error analysis of the model is pretty detailed and well-categorized. This will help future research in this area to solve these challenges.

Weakness:
* Please consider adding some details about the specific GPT-3 model being used here (curie) in the appendix. At the very least, cite the related papers/websites when describing baselines around lines 190-199.
* It'd help to have some examples in the main text about the difference between using stepwise and stepwise++ for decoding. That would make these sections more accessible to new readers.

---

> ### Author Response · Authors · 2022-08-01
> **Response to Reviewer 3**
>
> Thank you for your review and the comments on the presentation, comprehensive experiments, and error analysis. Please see our responses below.
>
> Re: "details about the specific GPT-3 model" and "cite the related papers/websites when describing baselines around lines 190-199".
> - Thanks for pointing this out, we will add citations, as well as further details about the models in Appendix F.
>
> Re: "It'd help to have some examples in the main text about the difference between using stepwise and stepwise++ for decoding"
> - That’s a great suggestion; we were low on space, but with the extra page allowed for the camera ready, we will add more explanation of the difference between stepwise vs. stepwise++ (expanding with multiple temperatures and selecting with multiple value weights).
>
> Re: "What is the order of the training data shown to GPT3 during fine-tuning - between in-context and reference reconstruction?"
> - The order is randomly sampled - we assemble a dataset containing both reference-reconstruction and proof generation examples, and randomly shuffle the dataset. Also note that the model is fine-tuned for multiple epochs, so after the first epoch it will have seen all of the reference-reconstruction examples.

---

### Official Review · Reviewer_PnA9 · 2022-07-12

**Rating:** 7
**Confidence:** 3
**Soundness:** 3 good
**Presentation:** 4 excellent
**Contribution:** 3 good

**Summary:**

This paper presents, NaturalProver, a machine learning-based proof solver that leverages the augments large language models (e.g., GPT-3) with symbolic information so they can be utilized for more than just natural language processing. The authors demonstrate NaturalProver’s advances in two fundamental dimensions: (1) full mathematical proof generation and (2) next-step proof generation. While the full proof generation component is limited to only a small number of steps for proof generation (e.g., 5-6 steps) the next-step proof assistance, in theory, could potentially be applied to any proof, irrespective of the total proof step count. The authors claim the next-step assistant is correct (and useful) up to 40% of the time.

The authors compare NaturalProver against a GPT-3 baseline.


**Questions:**

Can the authors please try to explain the ways they prevented the weaknesses I have listed above (pasted below) in the evaluation method for the paper?

Because humans were used to evaluate the correctness of the system, it seems as though there may have been some bias in the results for numerous reasons. First, if the students were aware which proofs were generated by the research team’s system, they may have intentionally (or unintentionally) scored such proofs higher. Second, because the graders are human – and of varying degrees of mathematical competence – there is some chance the grading for some proofs were incorrect. Third, according to the authors, the students were first shown a “golden proof” which is used as a key for the synthesized proofs. However, there can exist many different correct proofs for mathematical problems. Because of this, it could be the case that some of the generated proofs are correct but were flagged is incorrect because they deviated from the “golden proof” steps or solution.


**Limitations:**

I'm not aware of any potential negative societal impact this research could incur.


**Strengths And Weaknesses:**

Strengthens:

This paper is well-written and well-motivated.

The research topic is both timely (e.g., leveraging large language models for more than just natural language processing) and valuable (e.g., assisting formal solvers in ways that may improve some of the intractable computational bounds that have historically limited their capabilities to only relatively small proofs to do exponential state space explosion).

The two-pronged approach of both full proof and, perhaps more interestingly, next-step proof assistance, is compelling and has a reasonable amount of immediate value both in academia and industry (in my opinion).

Augmenting large language models to handle mathematical proof texts for the authors two-pronged approach is, to my knowledge, novel and clever. Additionally, the fact that the authors added LaTex support for mathematical proofs seems to imply it may have immediate value to other mathematicians in the research community, many of which write their mathematical proofs entirely in LaTex (myself included).

Weaknesses:

While GPT-3 is obviously one of the most well-known and effective LLMs to date, it would have been nice to see a comparison to other language models.

Because humans were used to evaluate the correctness of the system, it seems as though there may have been some bias in the results for numerous reasons. First, if the students were aware which proofs were generated by the research team’s system, they may have intentionally (or unintentionally) scored such proofs higher. Second, because the graders are human – and of varying degrees of mathematical competence – there is some chance the grading for some proofs were incorrect. Third, according to the authors, the students were first shown a “golden proof” which is used as a key for the synthesized proofs. However, there can exist many different correct proofs for mathematical problems. Because of this, it could be the case that some of the generated proofs are correct but were flagged is incorrect because they deviated from the “golden proof” steps or solution.

Minor nits:

Some of the written language in the paper seems a bit misplaced and perhaps in appropriate for a tier-1 research venue. This should hopefully be easy to correct, but I felt it worthwhile to mention. In the abstract there is mention that the LLMs generate “proofs riddled with hallucinations” – I have no idea what this means, but it seems like an inappropriate and unprofessional classification of results in a scientific research paper. In Section 5 under “human evaluation setup” the authors claim that “The human annotators were trained and compensated accordingly.” I have no idea what “compensated accordingly” means nor how to evaluate the accuracy of such a claim. Best to simply eliminate it entirely. I suspect it is fine to say that “students were trained on how to evaluate proof correctness.”

---

> ### Author Response · Authors · 2022-08-01
> **Response to Reviewer 2**
>
> Thank you for your detailed review, and we appreciate the comments on the novelty and value of our method and problem setting. Please see our responses related to evaluation and your other questions below.
>
> Re: "it would have been nice to see a comparison to other language models."
> - We do have a comparison with GPT-J 6B, GPT-2, and GPT-Neo (see the Language model comparison paragraph in 5.2). We would also like to highlight that we will provide the code and trained GPT-J/2 models as open source so that the community can further build upon and use them.
>
> Re: "if the students were aware which proofs were generated by the research team’s system, they may have intentionally (or unintentionally) scored such proofs higher"
> - To clarify, the annotators were _*not*_ aware of any properties of the system that generated the proofs; they were only told that the proof was computer generated (see Figure 3 for the interface).
>
> Re: "because the graders are human – and of varying degrees of mathematical competence – there is some chance the grading for some proofs were incorrect."
> - We acknowledge that evaluating proofs is a difficult task, so we took several steps to reduce the chance of incorrect evaluations impacting the results:
> 1. We designed the step-level and full-proof level annotation scheme along with training materials, and held a training session with the annotators so that they understood the evaluation setup.
> 2. We hired annotators among undergraduate students (or higher) of the math department, and placed a minimum course requirement to guarantee that they meet the qualifications.
> 3. We hired 15 annotators so that the human evaluation results do not depend on a single annotator’s evaluations.
> 4. We asked the annotators to only evaluate a proof if they understand it.
> 5. Finally, since the system is not known to the annotator, we would expect incorrect grading errors to be distributed across the compared systems.
>
> Re: possibility of proofs being flagged as incorrect because they deviated from the golden proof steps or solution.
> - We do not suspect that showing a gold proof had a substantial bias on our findings, since we additionally have annotators mark step level _errors_, which identify _why the proof is wrong_. The baseline GPT-3 has much higher error rates for various reasoning-related criteria, (Reasoning Errs, Table 2), which tells us that the annotators frequently identified specific aspects of the proof steps that were wrong. If the difference between systems was only due to the fact that the system’s proof differed from the ground-truth, we would not expect to see such a large difference in these errors.
>
> Re: written language
>
> - Thank you for the feedback on language, we’ve fixed the wording in the revision. We would be glad to remove or reword other cases. To clarify:
> 1. Regarding hallucinations, this was referring to the high rate of non-existent references generated by GPT-3 (e.g. see Table 3 for an example, and the “Halluc” metric in Table 6). We removed this statement from the abstract.
> 2. Regarding “compensated accordingly”, our study was approved by an Institutional Review Board that has minimum payment requirements, and we have the specific payment details in Appendix G.2. We adjusted the statement based on your feedback.

---

> > ### Comment · Reviewer_PnA9 · 2022-08-07
> > **Thank you!**
> >
> > Dear authors -
> >
> > Thank you for the explanations and addressing my concerns. My apologies for misunderstanding some things and thank you for the clarifications.
> >
> > Your response has addressed all of my questions and concerns. I have raised my score to accept.

---

### Official Review · Reviewer_YcPJ · 2022-07-18

**Rating:** 7
**Confidence:** 4
**Soundness:** 3 good
**Presentation:** 4 excellent
**Contribution:** 4 excellent

**Summary:**

This paper explores the prospect of theorem proving in natural mathematical language using large language models (LLMs). Two representative tasks are considered: generating the next step in a partial proof, and generating full proofs given only the statement of a theorem. A dataset for these tasks (based on the existing *NaturalProofs* benchmark) is contributed.

Unfortunately, naive attempts to fine-tune GPT-3 on these tasks are shown to fail -- generated proofs often contain "hallucinated" references or logical errors. To address this, the work proposes "NaturalProver" to enforce the inclusion of certain background references in the output. Empirically, NaturalProver is shown to outperform GPT-3 on a number of metrics, including usefulness and correctness (as determined by human evaluators).


**Questions:**

1. Would a version of constrained decoding but with soft (rather than hard) constraints be well-motivated? To me, it seems like the stepwise beam search can almost be applied as-is to this other setting (or if not, please correct me).

2. Is there a way to change NaturalProver to provide a measure of confidence in the correctness of the proof? This is arguably a key advantage of formal proof environments that's lost when dealing with natural language proofs.

**Limitations:**

Yes, the limitations and societal impacts were addressed.

**Strengths And Weaknesses:**

**Originality**: Recent research explores theorem proving with inputs in the form of natural mathematical language, including the paper that proposed the NaturalProofs dataset [1]. As the authors note, these works often focus on auxiliary tasks, such as reference retrieval or translation to a formal mathematical representation. This work, on the other hand, considers the ambitious (but important) task of generating proofs and proof steps end-to-end, *from* natural math language *to* natural math language. This is a novel problem setting for theorem proving, to my knowledge.

**Quality**:

[Method]
The proposed method motivates the separation of the problem into two parts: reference retrieval (either by a neural network, or a human) followed by proof generation, conditioned upon these references. This is well-justified, especially given how GPT-3 is shown to fail on the end-to-end task.

One assumption that I did not agree with was that the relevant references would always appear in the final proof. This assumption holds in the experiments when the ground-truth set of references can be provided, but is too strong for practical uses. For unclear reasons, the proposed approach is not evaluated with references retrieved by a neural network. (Please also see question 1 below).

[Experiments]
The various approaches are evaluated by human annotators on criteria that would otherwise be difficult to objectively measure. Automated metrics are proposed which correlate well with these criteria (with one exception). **I have a couple of concerns, which if addressed well would raise my score.**

- GPT-3 with beam search decoding should be a baseline to fairly compare against NaturalProver++.
- Error bars are missing on the main results, making it difficult to evaluate their significance.

**Clarity**:
The paper is very well-written and clear. One part I could not understand was the meaning of the "In-context" column in Table 4. Does "not in-context" mean no references were conditioned upon, similar to the vanilla GPT-3 baseline?

**Significance**:
End-to-end theorem proving in natural mathematical language is a highly significant problem. This paper presents many interesting and useful insights on the problem, including why naive applications of LLMs fail and challenges for the future. One concern that may discourage future work is the lack of objective measures of success, but this work also takes steps to address this by proposing automated metrics of proof quality. One limitation to the main contributed approach (the one that uses constrained decoding) is that it was only shown to succeed when provided with ground-truth references.

[1]: Welleck, Sean, et al. "NaturalProofs: Mathematical Theorem Proving in Natural Language." Thirty-fifth Conference on Neural Information Processing Systems Datasets and Benchmarks Track (Round 1). 2021.

---

> ### Author Response · Authors · 2022-08-01
> **Response to Reviewer 1**
>
> Thank you for your review and positive comments on the well-justified method, interesting and useful insights, and the significant and novel problem setting. Please see our responses related to retrieved references, constrained decoding, and your other questions.
>
> Re: "Assumption that relevant references would always appear in the final proof" and "the proposed approach is not evaluated with references retrieved by a neural network"
> - To clarify, we do evaluate with references retrieved by a neural network (NaturalProver-Retrieved), which uses our proposed reference-reconstruction & reference-conditioning to let the network learn to put relevant references into the proof. However, since the retrieved references typically include noise (violating the assumption), we don’t add in hard decoding constraints on the retrieved references.
>
> Re: "GPT-3 with beam search decoding should be a baseline to fairly compare against NaturalProver++."
> - Unfortunately token-level beam search is not available through the GPT-3 API. To address your concern, we’ve run an additional **stepwise stochastic beam search baseline**, which corresponds to using stepwise decoding with an LM-only value function (i.e. $\alpha=0.0$). Here are the results on the core validation set, with NaturalProver as the underlying model:
> |                          | Gleu | F1    | KF1   | Ref-P | Ref-R | Ref-F1 | Halluc |
> |--------------------------|------|-------|-------|-------|-------|--------|--------|
> | Stepwise Stochastic Beam | 41.0 | 68.89 | 90.33 | 91.43 | 82.04 | 84.21  | 4.6    |
> | NaturalProver++   | 40.4 | 68.90 | 97.24 | 95.05 | 94.85 | 94.15  | 2.0    |
>
> - As seen in the table, the constrained stepwise++ decoding used in NaturalProver++ substantially improves grounding metrics compared to stochastic beam search, while keeping the lexical content metrics at a similar level. We added this baseline into the revision (Appendix Table 13), and we will add a discussion into the main text given the extra page for the camera-ready.
>
> Re: Error bars
> - Due to the cost of human evaluation we do not have error bars for the main human evaluation results, but we do show standard deviation across 3 runs for the decoding algorithm ablations (Table 9 and 10).
>
> Re: "Does "not in-context" mean no references were conditioned upon, similar to the vanilla GPT-3 baseline?"
> - Correct - the first two rows show the effect of constrained decoding without conditioning on references (i.e. the references are only enforced via constraints).
>
> Re: "Would a version of constrained decoding but with soft (rather than hard) constraints be well-motivated?"
> - Yes, our stepwise++ constrained decoding supports any value function that is a function of the preceding context, so one could design a function based on soft constraints. However, doing so for the proof setting is nontrivial; for instance, if we define a soft constraint using an off-the-shelf sentence embedding similarity, two theorem titles might be “close”, yet only one is correct for the proof. Due to these hard logical requirements we focused on hard lexical constraints. However, developing alternative value functions is interesting future work.
>
> Re: "Is there a way to change NaturalProver to provide a measure of confidence in the correctness of the proof? This is arguably a key advantage of formal proof environments that's lost when dealing with natural language proofs."
> - Indeed verifiability is a key difference between formal and informal proofs, so developing confidence measures, either from NaturalProver itself or via an separate model, is an interesting direction for future research. Since the model’s log probabilities can be unreliable (e.g. repetitions are often assigned high scores), and since it is not obvious how to collect good positives/negatives to train a separate model, we consider this as a direction for future work rather than a modification to NaturalProver.

---

> > ### Comment · Reviewer_YcPJ · 2022-08-07
> > **Thanks for the response**
> >
> > Thank you for responding to my comments. Most of my concerns were adequately addressed and I will raise the score to "Accept".

---

### Author Response · Authors · 2022-08-01
**Summary Response**

We thank the reviewers for their thoughtful and encouraging comments.
The reviewers found NaturalProver to be a well-justified (R1), novel and clever method (R2), with comprehensive experiments and detailed error analysis (R3), and many interesting and useful insights (R1).
Moreover, the reviewers found the problem of generating proofs and proof steps end-to-end in natural mathematical language to be timely and valuable (R2), as well as highly significant and novel (R1).

We have addressed the concerns and feedback from each reviewer below, including the corresponding adjustments that we will make in the revision.

---

### Meta-Review · Area_Chair_teiZ · 2022-08-21

**Recommendation:** Accept
**Confidence:** Certain

**Metareview:**

The paper addresses an exciting problem statement--generating theorems directly in natural language--and shows how to adapt large language models to this task, both for autocompletion, proof reference generation, and wholecloth proof generation. While previous works have considered various auxiliary mathematical tasks posed in natural language, this work takes an important step by making progress toward doing proofs directly in natural language. This is a hard problem, and the authors support their work with experiments showcasing and analyzing different kinds of successes and failures. The reviews are unanimous in recommending acceptance.

**Award:**

No

---

### Decision · Program_Chairs · 2022-09-14

Accept